# Investigating the Role of Urban Vehicle Access Regulations as a Policy Tool for Promoting Electric Mobility in Budapest

Gabriel Ayobami Ogunkunbi and Ferenc Meszaros *

Department of Transport Technology and Economics, Faculty of Transportation Engineering and Vehicle Engineering, Budapest University of Technology and Economics, Műegyetem rkp. 3., H-1111 Budapest, Hungary
* Correspondence: meszaros.ferenc@kjk.bme.hu

**Abstract:** To promote sustainable urban mobility and reduce environmental pollution, transportation policies worldwide aim to decrease reliance on fossil fuels. This requires reducing private car use through policy instruments such as urban vehicle access regulations (UVARs) and promoting alternative sustainable transport technologies such as electromobility. Considering that the deployment of such regulations and the market penetration of battery electric vehicles (BEVs) is still low in Hungary, this study aimed to examine the willingness of urban dwellers in Budapest, Hungary, to adopt battery electric vehicles (BEVs) upon implementation of an UVAR measure. The study analysed the BEV adoption intention of 409 urban residents who participated in an UVAR study in 2022. The results show that age is a significant factor, with individuals aged 35–44 most likely to adopt BEVs. However, other socio-demographic characteristics did not significantly affect willingness to adopt BEVs. Additionally, pro-environmental behaviour or attitude did not significantly predict BEV adoption. Based on these findings, this study highlights the importance of considering multiple interrelated factors and provides policy insights for promoting sustainable transportation technology adoption.

**Keywords:** transport decarbonisation; zero-emission vehicles; electromobility; demand management; urban mobility; Budapest

## 1. Introduction

Transport decarbonisation is a crucial challenge facing the world as it seeks to mitigate the impacts of climate change. The transportation sector contributes significantly to global greenhouse gas emissions with road vehicles, particularly passenger cars, being the primary contributors due to their heavy reliance on fossil fuels [1]. Despite a 24% decrease in greenhouse gas emissions in 2019 compared to 1990, decarbonising the transport sector in the European Union has been slow with emissions from the sector still rising [2]. The importance of continued and more ambitious efforts to decarbonise the transport sector to achieve the goals of the Paris Agreement cannot be overemphasised. Reducing emissions in the transport sector by 3% annually until 2030 is critical to align with the Net Zero Emissions by 2050 Scenario [3]. Achieving this will require innovative technology, changes in consumer behaviour, and policy interventions at the local, national, and international levels. Addressing these concerns and other negative impacts of transport, including air pollution, noise, congestion, and safety concerns, will be a significant step in the right direction.

To reduce negative impacts, strategies that combine technological changes, changes in consumer behaviour, and policy motivations are often organised using the avoid–shift–improve framework [4]. The framework provides a clear structuring approach to improve transport sustainability. The framework separates policy interventions into three categories: avoid, shift, and improve [5,6]. Avoid policies aim to reduce the need for travel, shift policies aim to change the mode of travel, and improve policies aim to increase the energy efficiency of the transportation system [7–9].

The growing focus on electromobility as a means of building sustainable transport systems in urban areas is reflected in its increasing prominence in EU transport policy and the policies of individual member states. Electromobility is considered a critical tool in limiting the environmental impact of transportation [10]. Battery electric vehicles (BEVs) have been identified as a promising alternative to internal combustion engine vehicles due to their innovative technology. The innovative technology can disrupt the traditional passenger car regime [11,12]. BEVs, therefore, play a key role in transport decarbonisation and sustainable mobility transition [13]. Despite the potential benefits of BEVs, their low market uptake has limited their gains. This can be attributed to several factors, including the relatively high cost of BEVs compared to traditional internal combustion engine vehicles, the limited range of many BEVs, and the lack of charging infrastructure in many areas [14,15]. Realising the potential benefits of BEVs requires incentivising their uptake through government subsidies for the purchase of BEVs, tax waivers, or deductions for BEV adopters as well as investment in charging infrastructure. However, the burden of these subsidies on government finances implies they cannot be sustained long-term and would decrease and eventually fade out [16]. This has led to different studies exploring the impact of non-financial or alternative incentive policies (e.g., [17–20]).

Urban vehicle access regulation (UVAR) is one alternative policy intervention widely adopted in Europe (see [21]), which has the potential to boost the market adoption of BEVs. In addition, UVAR has been proven to be an effective intervention to reduce car dependency [22]. Depending on design and perspective, they may have elements of avoid, shift, and improve policy interventions [23,24]. Concerning BEVs, UVAR, which raises the cost of conventionally-fuelled vehicle usage, can incentivise BEV uptake while reducing the number of unnecessary trips. This shift in mobility behaviour can also be seen as an improvement as it reduces emissions from the transportation sector and further stimulates technological development.

Studies have investigated the influence of different forms of UVAR on electric vehicle adoption. An investigation of the impact of the London congestion charge on the registration rate of hybrid electric vehicles (HEV), which were exempted from the road charges, found that proximity to the congestion charging area is positively associated with HEV registrations [20]. Bjerkan and colleagues surveyed BEV owners in Norway to determine incentives that motivated their purchase decisions, and toll fee exemption was one of the influential policy incentives [25]. Using numerical analysis, a study on BEV exemption from traffic restrictions in China found that exemptions promote customer adoption [19]. However, another study in China using a stated preference approach did not find a significant influence of driving restrictions and congestion charges on electric vehicle adoption [26].

Against this backdrop, this study aims to contribute to the discussion on the influence of access regulation implementation on BEV adoption. It builds on data collected in a stated preference survey on the willingness to support UVAR measures conducted in Budapest, Hungary. It proposes to answer the following research questions: What factors might influence urban residents to switch to a BEV if an UVAR is implemented? What categories of urban commuters will be willing to shift to BEVs within the same context? This study's contribution lies in its examination of the potential role of UVAR on the market adoption of BEVs and the effects of socio-demographic, transport-perception, and trip-behaviour factors. It adds to the body of knowledge of studies that discuss the driving and limiting factors of BEV adoption at the individual level examined from an UVAR planning perspective. While a wide range of BEVs are used within urban areas, including electric personal mobility vehicles (e.g., bicycles, scooters, and segways), passenger cars, and buses, this study focuses on battery electric passenger cars. By answering these research questions, this study provides insights into the effectiveness of UVAR in boosting BEV market share and the role of different factors in influencing BEV uptake. This information will be useful for policymakers and stakeholders seeking to develop effective policies to promote the uptake of low-carbon vehicles.

## 2. Geographical and Policy Context

Budapest, the capital city of Hungary, functions as an important economic, logistical, and cultural centre of the Central European country. The city is the most densely populated in the country with a population density of 3204 people per square kilometre, and it accounts for 17% of the country's total population [27]. Significant population growth in Budapest and its agglomeration areas has increased the demand for personal mobility. This growth has also contributed to urban sprawl as the city has expanded outwards, leading to longer travel distances and increased car dependency. Consequently, like many similar cities, Budapest faces significant mobility challenges, including traffic congestion, air pollution, and noise pollution, which negatively impact the city's environment, economy, and society.

To address these challenges and promote sustainable urban mobility, national and metropolitan authorities have implemented various policies and initiatives, including promoting the use of electric cars and planned measures of regulating access to conventionally fuelled vehicles. Hungary adopted a revised National Electromobility Development strategy, the Jedlik Ányos Plan 2.0, in 2019. The strategy aims to establish a national model for the electromobility market to ensure that electric vehicles are accessible to the widest range of retail and institutional customers. Additionally, the plan includes expanding charging infrastructure, promoting the use of electric vehicles, and a commitment by central and local authorities to roll out charging infrastructure and fleet expansion using electric vehicles [28]. Consequently, several end-user-focused financial support schemes have been implemented, including purchase subsidies and other tax incentives. In addition, non-financial incentives are in place for alternatively fuelled vehicles, including exemption from vehicle prohibitions during smog alerts, exemption from parking fees, and access to limited traffic zones in historical urban centres. However, these non-financial incentives are discretionary measures provided at the municipality level.

Owing to these efforts, the market share of electric vehicles (including BEVs and PHEVs) has increased from 0.13% of the total vehicle fleet in 2017 to about 1.06% in 2021. 43% of this EV fleet is registered in Budapest [29]. While the growth rate might be low compared to other European countries, it is among the highest within the Central and Eastern European bloc [30]. Regardless, Hungary's current incentives are still inadequate [29], particularly since the country could only achieve about 50% of its 21,000 indicative electric passenger cars fleet target in the erstwhile National Policy Framework for the Development of Alternative Fuels Infrastructure [31,32]. Yet, Hungary has a more ambitious mid-term target of about 300,000 EVs in use by 2025 in the Jedlik Ányos Plan 2.0 [28].

On another front of inducing sustainable mobility behaviour amongst inhabitants and for climate change mitigation, Budapest included an emission-proportionate congestion charging scheme in all planning scenarios of its sustainable urban mobility plan (SUMP) [33]. It also listed a low emission zone as a potential measure amongst other planned measures in its climate action plan [34]. These two planned urban vehicle access regulation measures present an opportunity for Budapest to further incentivise the adoption of battery electric vehicles (BEVs) if these vehicles are exempted from the proposed measures. Implementing these regulations may significantly impact BEV market adoption as they can serve as a strong incentive for potential buyers. However, the effectiveness of these measures in motivating urban dwellers to switch to BEVs remains to be seen.

## 3. Methods

### 3.1. Data Collection and Variables

This study sets out to investigate the influence implementation that UVAR might have on BEV adoption. It uses data from a survey of urban residents' preferences for urban vehicle access regulations conducted by the authors between May and July 2022 in Budapest, Hungary. The survey, made available in Hungarian and English, took the form of online self-administered questionnaires hosted on Sawtooth Software servers. This approach allowed respondents to self-select for participation in the study. It further ensured the

participants had ample time to complete the survey and guaranteed anonymity. Questions and question groups were structured to identify current travel behaviour, urban traffic concerns, disposition to vehicle regulation implementation, and the socio-demographic characteristics of respondents. For most of the questions, five-point Likert-type scale response options were presented.

For this research, the survey item, "I will switch to or buy a fully electric car (BEVs) if the measures (UVAR) are implemented", answered on a 5-point agreement scale, was the variable of interest. To characterise the respondents who are willing to adopt a BEV upon the implementation of the planned UVAR in Budapest and to identify the factors influencing their choices, we included the following socio-demographic characteristics as independent variables in the analysis: gender (male/female), age (18–34; 35–44; 45–54; and greater than 55), educational level (secondary education or less; first degree; and higher degrees), employment status (having a paid job/not having a paid job), income categories, mode of daily commute (passenger car; public transport; walking; micromobility modes; and other), possession of valid driving license (yes/no), and typical commute trip origin and destination (near or around city centre/outside the city centre). Other variables included factors considered in trip planning (time; cost; and environment), perceived dissatisfaction with the main urban transport modes (passenger car; public transport; cycling; and walking), awareness of transport problems (air pollution; noise annoyance, congestion; parking; public transport; active mobility; and safety) and willingness to support car-free urban areas (yes; no; and indifferent).

### 3.2. Analysis

The data were analysed using IBM SPSS 29 [35]. A descriptive bivariate analysis was conducted to understand and characterise the sample distribution. A Chi-square test was also conducted between the independent and dependent variables, particularly to compare differences across the response categories of the dependent variable. The bivariate analysis also allowed us to ascertain that the data did not violate some of the assumptions of the planned modelling approach.

Since the level of agreement regarding adoption of a BEV if UVAR is implemented is categorical and can be assumed to have a natural ordering, the ordinal logistic regression or multinomial logistic regression is appropriate to explore the factors influencing respondents' decisions. However, tests of parallel lines suggested that the important proportional odds assumption was violated in the ordinal logistic regression [36,37]. Hence, we adopted the multinomial logistic regression. Furthermore, for model simplicity and to reduce statistical errors, the dependent variable was recoded into three categories: "Yes" (somewhat agree and strongly agree responses), "No" (somewhat disagree and strongly disagree responses), and "Neutral" (neither disagree nor agree responses). Model simplification is encouraged as model interpretation can be difficult with more than four dependent variable categories [38]. We conducted diagnostic tests, including a check of the model assumptions using variance inflation factor (VIF) tests to detect multicollinearity. A VIF greater than 5.0 indicates a high risk of multicollinearity [39]. Our model did not show a risk of multicollinearity as the maximum value of the VIF test statistic across all variables was 2.8.

The multinomial logistic regression estimates k-1 logit models, with k representing the number of dependent variable categories and one category designated as the reference group. The model uses maximum likelihood estimation to evaluate the odds of categorical membership relative to the reference group in the logit model [40]. We specified a multinomial logistic regression model to estimate the probability of adopting an EV as the next vehicle with "No" as the reference category. We included main effects for all independent variables. To enhance the efficiency of the estimators, we used the backward stepwise regression technique to eliminate the explanatory variables that were significantly less relevant. Explanatory variables with $p$-values greater than 0.30 were excluded from the model. We adopted this liberal selection criterion as excessively stringent threshold values such as $p > 0.05$ could exclude important predictors due to a lack of statistical power [38,41].

This also explains why we did not estimate the model using only variables significantly associated with the dependent variable based on the Chi-square test output.

## 4. Results

### 4.1. Descriptive Statistics

The survey received 409 valid responses. While the sample size can be considered relatively low, it satisfies the rule of thumb of more than ten events per variable often required for logistic regression [42,43]. In addition, some variable categories were merged (age groups 18–25 with 25–34 and income groups HUF 400,000–600,000 with greater than HUF 600,000) to reduce the risk of biased estimated odd ratios and model overfitting associated with lower events per variable [44,45]. The summary of the background characteristics of the analytic sample is presented in Table 1.

Overall, there was a balanced representation of gender and locations of typical trip origin. There was, however, an overrepresentation of respondents who use public transport as the commuting mode of travel (52.1%), those who have a paid job (75.8%), and the median income group (52.6%). The proportion of respondents who own a valid driving license (72.0%) also far outweighs the underrepresented proportion of those who use passenger cars typically for their daily commute (25.4%), indicating less dominance of passenger car travel regardless of the driving ability of most respondents. Benchmarking with the population data of Budapest, the survey sample can be said to be representative only across gender. The sample distribution is also marginally similar to the city's employment level.

The distribution of the categories of the willingness to adopt BEVs across the background variables is presented in Table 2. The analysis of the frequencies shows that respondents in the age group 35–44, who have only attained a first degree, are gainfully employed and belong to the highest income segment, are more willing to adopt BEVs if the planned Budapest UVAR measures are implemented. Likewise, those who commute via public transport or micromobility modes (predominantly bicycles and scooters), are likely without valid driver's licenses, and have trips originating or ending around the inner-city core are potential BEV adopters. Expectedly, the greater proportion of respondents who were willing to support or who were indifferent about car-free measures agreed to shift to BEVs. However, of the presented variables, only commuting mode, valid driver's license, and willingness to support UVAR measures were significantly associated with the dependent variable based on the Chi-square test results.

Considering the perceived dissatisfaction with the four prevalent urban transport modes, the distribution across the dependent variable is visualised in Figure 1. Generally, most respondents disagreed with the constructs, stating they were dissatisfied with all urban transport modes except passenger cars. Respondents agreed that they were dissatisfied with using passenger cars for their travels. The distribution across the "Yes", "Neutral", and "No" responses also had similar patterns with no significant difference. However, perceived dissatisfaction with cycling was found to be significantly associated with the choice to adopt BEVs. Most respondents who stated they would adopt BEVs neither agreed nor disagreed with being dissatisfied with cycling in Budapest.

Time and cost are the most important factors the respondents consider while planning their trips, while only a minority take the environment into consideration. The frequencies of the responses across the dependent variable are shown in Figure 2. Surprisingly, fewer people who agreed or strongly agreed to be environment-conscious trip planners would be willing to adopt BEVs if UVAR is implemented in Budapest compared to those who prioritise time and cost.

**Table 1.** Background characteristics of study participants.

| Characteristic | Sample Frequency (%) | Population Data [a] |
|---|---|---|
| **Gender** | | |
| Female | 221 (54.0%) | 53.0% |
| Male | 188 (46.0%) | 47.0% |
| **Age** | | |
| 18–34 | 115 (28.1%) | 24.3% |
| 35–44 | 81 (19.8%) | 17.3% |
| 45–54 | 111 (27.1%) | 19.0% |
| 55 or older | 102 (24.9%) | 39.3% |
| **Educational attainment** | | |
| Secondary or less | 196 (47.9%) | |
| First degree | 115 (28.1%) | |
| Higher degree | 98 (24.0%) | |
| **Employment status [b]** | | |
| Having a paid job | 310 (75.8%) | 69.9% |
| Not having a paid job | 99 (24.2%) | 30.1% |
| **Income [c]** | | |
| Less than HUF 200,000 | 112 (27.4%) | |
| HUF 200,000–HUF 400,000 | 215 (52.6%) | |
| Greater than HUF 400,000 | 82 (20.0%) | |
| **Commuting mode [d]** | | |
| Passenger Car | 93 (22.7%) | Private car: 35% |
| Public Transport | 213 (52.1%) | Public transport: 47% |
| Walking | 29 (7.1%) | Walking: 16% |
| Micromobility (cycling and scooters) | 32 (7.8%) | Cycling: 2% |
| Other | 42 (10.3%) | - |
| **Typical trip origin** | | |
| Near or around the city centre | 224 (54.8%) | |
| Outside the city centre | 185 (45.2%) | |
| **Typical trip destination** | | |
| Near or around the city centre | 223 (54.5%) | |
| Outside the city centre | 186 (45.5%) | |
| **Valid driver's license** | | |
| Yes | 280 (68.5%) | |
| No | 129 (31.5%) | |
| **Willingness to support UVAR** | | |
| Yes | 175 (42.8%) | |
| No | 141 (34.5%) | |
| Indifferent | 93 (22.7%) | |
| **Intention to adopt BEV** | | |
| Yes | 159 (38.9%) | |
| Neutral | 98 (23.9%) | |
| No | 152 (37.2%) | |

[a] All population data were sourced from the Hungarian Central Statistical Office [46] except "Commuting mode". Numbers represent the Budapest population data except "Age", which represents national data. The data were enumerated into age groups by the authors. [b] Population data was the employment rate for the population aged 15–74. [c] Population's income data was only provided in quintiles. The average monthly gross income per capita for the year 2021 is estimated at HUF 265,000. [d] Data from the 2021 modal split survey in Budapest [47].

**Table 2.** Association between background characteristics of the study and intention for BEV adoption.

| Characteristic | Willingness to Adopt BEV | | | Chi-Square |
|---|---|---|---|---|
| | **Yes (%)** | **Neutral (%)** | **No (%)** | |
| Gender | | | | |
| Female | 39.4 | 23.5 | 37.1 | 0.068 |
| Male | 38.3 | 24.5 | 37.2 | |
| Age | | | | |
| 18–34 | 35.7 | 26.1 | 38.3 | |
| 35–44 | 50.6 | | 23.5 | |
| 45–54 | 37.8 | | 40.5 | 9.895 |
| 55 or older | 34.3 | | 43.1 | |
| Educational attainment | | | | |
| Secondary or less | 33.7 | 29.1 | 37.2 | |
| First degree | 46.1 | 20.0 | 33.9 | 7.854 |
| Higher degree | 40.8 | 18.4 | 40.8 | |
| Employment status | | | | |
| Having a paid job | 40.0 | 22.9 | 37.1 | 1.016 |
| Not having a paid job | 35.4 | | 37.4 | |
| Income | | | | |
| Less than HUF 200,000 | 35.7 | 32.1 | 32.1 | |
| HUF 200,000–HUF 400,000 | 38.6 | 21.4 | 40.0 | 6.455 |
| Greater than HUF 400,000 | 43.9 | 19.5 | | |
| Commuting mode | | | | |
| Passenger Car | 38.7 | 15.1 | 46.2 | |
| Public Transport | 40.8 | 29.6 | 29.6 | |
| Walking | 24.1 | 20.7 | | 16.616 * |
| Micromobility modes | 40.6 | 21.9 | 37.5 | |
| Other | 38.1 | 19.0 | 42.9 | |
| Typical trip origin | | | | |
| Near or around the city centre | 39.3 | 27.2 | 33.5 | 4.039 |
| Outside the city centre | 38.4 | 20.0 | 41.6 | |
| Typical trip destination | | | | |
| Near or around the city centre | 38.6 | 26.0 | 35.4 | 1.269 |
| Outside the city centre | 28.0 | 21.5 | 39.2 | |
| Valid driver's license | | | | |
| Yes | 37.5 | 20.0 | 42.5 | 13.047 ** |
| No | 41.9 | | 25.6 | |
| Willingness to support UVAR | | | | |
| Yes | 44.6 | 27.4 | 28.0 | |
| No | 29.8 | 18.4 | 51.8 | 20.139 *** |
| Indifferent | 41.9 | 25.8 | 32.3 | |

* $p < 0.05$; ** $p < 0.01$; *** $p < 0.001$.

Respondents also rated their awareness of urban transport problems on a five-point Likert scale, ranging from "not concerned" to "extremely concerned". Results for this rating showed that air pollution, congestion, and parking were the greatest concerns. At the same time, issues with safety, public transport, and active mobility were rated to be of least concern. The distribution of the frequencies of these variables across the dependent variables is presented in Figure 3. Only congestion and parking showed a significant association with the decision to adopt BEVs. Most respondents deeply concerned with these transport concerns would be willing to shift to a BEV.

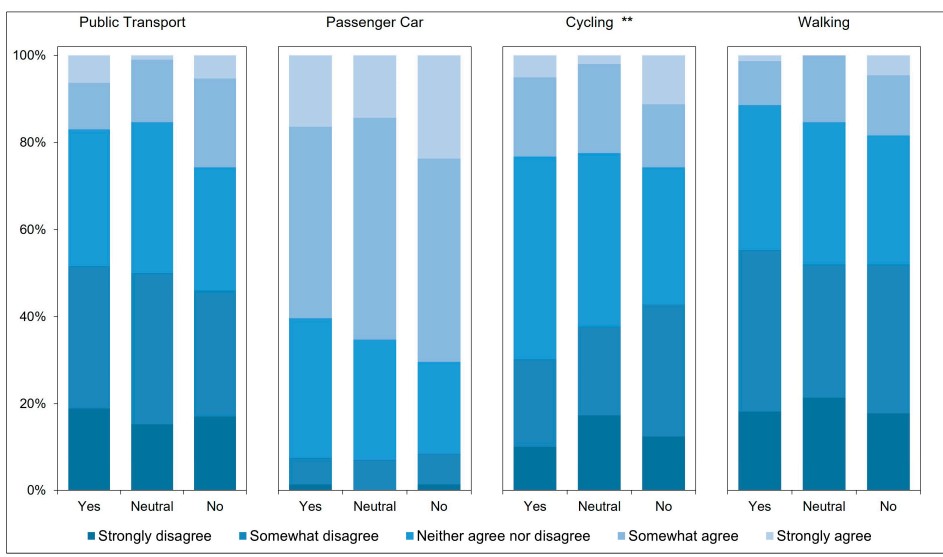

**Figure 1.** Respondents' perceived dissatisfaction with urban transport modes. ** *p* < 0.01.

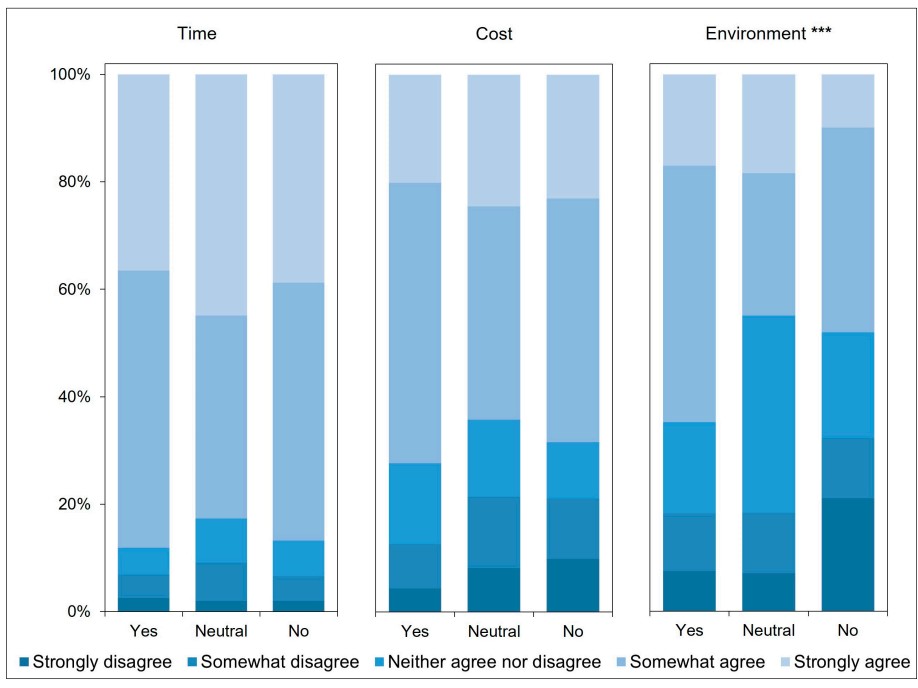

**Figure 2.** Respondents' identification of trip planning factors according to intention to adopt BEVs. *** *p* < 0.001.

Commuting mode and the possession of a valid driver's license were the only socio-demographic variables significantly associated with the intention to adopt an electric vehicle. Considering travel behaviour, attitude and problem awareness, variables representing environmentally conscious trip planning, and the awareness of congestion and parking problems were found to be significant. Lastly, the willingness to support UVAR was also identified to be significantly associated with the intention to adopt an electric vehicle.

There are many plausible explanations for these significant associations. For example, the higher proportion of people who do not have a driving license willing to adopt a BEV in an UVAR scenario may be attributed to their lack of personal investment in conventionally fuelled vehicles and the potential benefits BEVs will offer upon adoption. Similarly, the preference for BEV adoption amongst the early-middle age group may be influenced by

financial stability and the need for home–work commuting within or across the vehicle-regulated area. Traffic density reduction due to UVAR implementation may make adopting and driving BEVs more attractive for those concerned about congestion and parking. Pro-environmentalism is the most probable explanation for the significant association between people who factor environmental impacts into their trip planning decisions, people willing to support UVAR implementation, and the intention to adopt an electric vehicle.

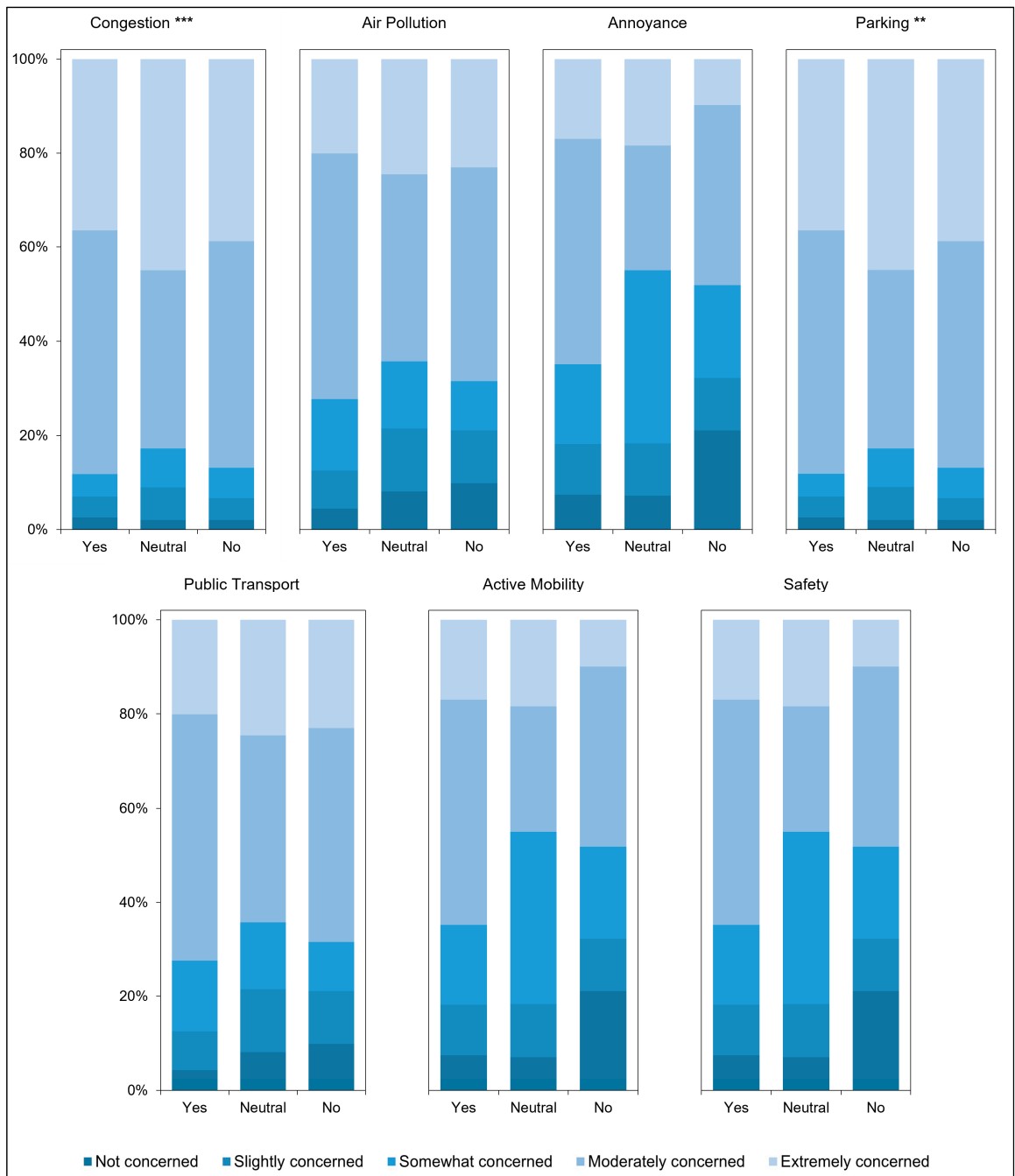

**Figure 3.** Respondents' awareness of urban transport problems according to intention to adopt BEVs. ** *p* < 0.01; *** *p* < 0.001.

### 4.2. Model Parameters and Estimates

The results of the multinomial logistic regression modelling performed to ascertain the effects of the various explanatory factors on BEV adoption intention are presented in

Table 3. The table shows two models for each non-negative response to adopting BEVs–Yes and Neutral. The models indicate the odds of agreeing to adopt BEVs or being neutral relative to disagreeing with adopting BEVs.

**Table 3.** Odds ratios and confidence intervals for factors affecting BEV adoption: results from multinomial logistic regression.

| Explanatory Variable | Yes | | | Neutral | | |
|---|---|---|---|---|---|---|
| | Odds Ratio | 95% CI | | Odds Ratio | 95% CI | |
| | | Lower | Upper | | Lower | Upper |
| Commuting Mode (Ref = Passenger Car) | | | | | | |
| Public Transport | 1.197 | 0.558 | 2.566 | 2.631 | 0.952 | 7.268 |
| Walking | 0.241 * | 0.068 | 0.857 | 0.630 | 0.130 | 3.044 |
| Micromobility modes | 1.309 | 0.407 | 4.218 | 1.264 | 0.282 | 5.658 |
| Other | 0.687 | 0.225 | 2.098 | 0.685 | 0.157 | 2.994 |
| Mode Dissatisfaction: Public Transport (ref = strongly agree) | | | | | | |
| Strongly disagree | 0.777 | 0.189 | 3.198 | 2.834 | 0.195 | 41.105 |
| Somewhat disagree | 0.685 | 0.185 | 2.535 | 4.059 | 0.309 | 53.254 |
| Neither agree nor disagree | 0.607 | 0.165 | 2.235 | 4.580 | 0.352 | 59.607 |
| Somewhat agree | 0.245 * | 0.061 | 0.982 | 1.894 | 0.139 | 25.838 |
| Mode Dissatisfaction: Cycling (ref = strongly agree) | | | | | | |
| Strongly disagree | 1.525 | 0.380 | 6.114 | 8.981 * | 1.105 | 73.004 |
| Somewhat disagree | 1.159 | 0.327 | 4.105 | 4.188 | 0.540 | 32.461 |
| Neither agree nor disagree | 2.965 | 0.881 | 9.974 | 8.720 * | 1.208 | 62.950 |
| Somewhat agree | 2.377 | 0.665 | 8.496 | 15.360 ** | 1.927 | 122.413 |
| Trip Planning: Cost (ref = strongly agree) | | | | | | |
| Strongly disagree | 1.008 | 0.267 | 3.808 | 2.573 | 0.559 | 11.851 |
| Somewhat disagree | 0.658 | 0.209 | 2.070 | 2.635 | 0.686 | 10.111 |
| Neither agree nor disagree | 3.104 * | 1.081 | 8.914 | 2.701 | 0.750 | 9.723 |
| Somewhat agree | 1.575 | 0.738 | 3.363 | 1.749 | 0.707 | 4.327 |
| Trip Planning: Environment (ref = strongly agree) | | | | | | |
| Strongly disagree | 0.308 * | 0.095 | 1.000 | 0.145 ** | 0.036 | 0.582 |
| Somewhat disagree | 0.654 | 0.200 | 2.135 | 0.257 | 0.064 | 1.034 |
| Neither agree nor disagree | 0.558 | 0.198 | 1.571 | 1.147 | 0.365 | 3.602 |
| Somewhat agree | 0.931 | 0.367 | 2.358 | 0.274 * | 0.089 | 0.837 |
| Problem Awareness: Congestion (ref = Extremely concerned) | | | | | | |
| Not concerned | 1.076 | 0.113 | 10.198 | 2.822 | 0.365 | 21.785 |
| Slightly concerned | 6.739 ** | 1.745 | 26.030 | 3.279 | 0.718 | 14.978 |
| Somewhat concerned | 2.147 | 0.820 | 5.623 | 0.690 | 0.219 | 2.174 |
| Moderately concerned | 1.622 | 0.748 | 3.519 | 0.454 | 0.176 | 1.171 |
| Problem Awareness: Parking (ref = Extremely concerned) | | | | | | |
| Not concerned | 0.520 | 0.110 | 2.457 | 2.199 | 0.410 | 11.781 |
| Slightly concerned | 0.806 | 0.246 | 2.644 | 1.104 | 0.253 | 4.814 |
| Somewhat concerned | 1.019 | 0.402 | 2.582 | 2.330 | 0.718 | 7.561 |
| Moderately concerned | 1.794 | 0.860 | 3.745 | 5.696 *** | 2.284 | 14.201 |
| Problem Awareness: Active Mobility (ref = Extremely concerned) | | | | | | |
| Not concerned | 0.730 | 0.189 | 2.818 | 0.586 | 0.124 | 2.765 |
| Slightly concerned | 0.400 | 0.112 | 1.425 | 0.255 | 0.056 | 1.170 |
| Somewhat concerned | 0.405 | 0.132 | 1.242 | 0.182 * | 0.047 | 0.708 |
| Moderately concerned | 0.674 | 0.205 | 2.220 | 0.332 | 0.084 | 1.311 |

**Table 3.** *Cont.*

| Explanatory Variable | Yes | | | Neutral | | |
|---|---|---|---|---|---|---|
| | Odds Ratio | 95% CI | | Odds Ratio | 95% CI | |
| | | Lower | Upper | | Lower | Upper |
| Problem Awareness: Safety (ref = Extremely concerned) | | | | | | |
| Not concerned | 0.431 | 0.102 | 1.825 | 0.687 | 0.108 | 4.381 |
| Slightly concerned | 1.058 | 0.367 | 3.047 | 0.847 | 0.221 | 3.246 |
| Somewhat concerned | 0.953 | 0.384 | 2.365 | 1.908 | 0.627 | 5.807 |
| Moderately concerned | 1.242 | 0.502 | 3.073 | 3.457 * | 1.124 | 10.632 |
| Age (ref = 55 or older) | | | | | | |
| 18–34 | 0.977 | 0.425 | 2.243 | 0.781 | 0.281 | 2.171 |
| 35–44 | 3.640 ** | 1.479 | 8.962 | 2.088 | 0.687 | 6.352 |
| 45–54 | 1.164 | 0.529 | 2.564 | 0.963 | 0.354 | 2.615 |
| Valid driver's license (ref = Yes) | | | | | | |
| No | 2.447 * | 1.189 | 5.038 | 2.216 | 0.962 | 5.105 |

The −2 Log Likelihood Intercept Only = 873.08, −2 Log Likelihood Final = 865.65, Model Chi-square = 227.42 significant at $p$ = <0.001, and Pseudo $R^2$ (McFadden) = 0.258. * $p < 0.05$; ** $p < 0.01$; *** $p < 0.001$.

The model provides an acceptable fit to the data. The logit model is significant at $p < 0.001$ with a great difference between the -2 log-likelihood of the intercept only and the final model, suggesting that the independent variables contribute significantly to the prediction of the dependent variable. The McFadden pseudo $R^2$ value (McFadden) of 0.258 is considered satisfactory as it lies between the recommended range of 0.2 to 0.4 [48]. Goodness-of-Fit statistics with the predictors in the model show a good fit with a non-significant test statistic according to the Deviance criterion ($p$ = 0.215). The model has an overall classification accuracy of 64.1%, accurately predicting 67.9% of those willing to adopt BEVs. A chance accuracy test based on the marginal frequencies of the dependent variable, as described by [49], produced a chance accuracy rate of 43.4%. The model has adequate accuracy with the prediction improving on chance by more than 25% (64.1% > 43.3%).

The identified transport-related factors that significantly predict the intention for urban dwellers to adopt BEVs when Budapest implements the planned UVAR measures include mode of commuting and perceived dissatisfaction with public transport and cycling. Other transport-related factors include cost-prioritising and environmentally conscious trip planning and the awareness of congestion, parking, active mobility, and safety problems.

Those who walk to their primary commute destinations are found to be less likely to make a switch to BEVs compared to passenger car users. In comparison to those who strongly agree, those who somewhat agree to be dissatisfied with public transport service in the city are less likely to be potential adopters. The model parameters related to cycling dissatisfaction are, however, fuzzy. Except for those who somewhat disagree with being dissatisfied, people responding with the remaining spectrum of opinions are likely to be neutral regarding BEV adoption. The reality that not everyone cycles within the urban area might explain this result.

The model also suggests that those uncertain whether they prioritise cost while planning trips are three times more likely to buy or change their vehicles to non-hybrid electric cars than those who strongly agree with prioritising cost. People who strongly differ regarding environmentally conscious trip planning are also less likely to adopt BEVs than those who are strongly pro-environment. Surprisingly, even those who somewhat agree to be pro-environment may not adopt fully electric vehicles.

People slightly concerned about the prevalence of congestion are more likely to shift to BEVs in a vehicle-access-regulated Budapest than those who are extremely concerned about congestion. In addition, identifying as only moderately concerned about parking

or safety-related problems in the city could indicate that a person is likely neutral about BEV adoption compared to those who are extremely concerned. Respondents who are moderately concerned with parking are also more likely to adopt BEVs than those who are not concerned. On the other hand, those who are somewhat concerned about the challenges of active mobility are less likely to be neutral about BEV adoption compared to those who are extremely concerned.

Age and valid driver's license possession are the only significant socio-demographic determinants of BEV adoption from the modelling results. Urban dwellers aged 35–44 are almost four times more likely to switch to a fully electric vehicle should an UVAR be implemented in Budapest than those older than 54. Counterintuitively, persons who do not have a valid driver's license–compared to those who have–are twice more likely to shift their mobility to BEVs.

## 5. Discussion

The frequency analysis of the responses to the study's variable of interest did not show a clear preference for BEV adoption within an UVAR scenario. There was only a marginal difference between the proportion of respondents who agreed to adopt BEVs and those who disagreed. However, the investigation of the factors that significantly characterised those who would be willing to adopt BEVs offers some insights.

Willingness to adopt BEVs increased with age until age 35–44. Beyond this age group, a decline in willingness to adopt BEVs was observed. This finding suggests that age plays a significant role in determining an individual's desire to adopt BEVs, with younger and middle-aged individuals being more likely to adopt the technology. This further extends similar findings from similar studies conducted in Germany and the United States [50,51]. Investigating other factors such as employment status, gender, income, and spatial trip pattern did not reveal any significant association with willingness to adopt BEVs. Our results, therefore, agree with studies that found socio-demographic variables to have low explanatory power, especially when combined with other factors (e.g., [52,53]). These results have important implications for policymakers and stakeholders in the EV industry. They suggest that age-specific strategies may be more effective in increasing BEV adoption than those considering the employment, income, and spatial location of residents in the study area.

An interesting finding from this study is the relationship between possessing a valid driver's license and the intention to adopt BEVs. On the one hand, the results suggest that non-drivers (those who do not have a valid driver's license) may be more willing to adopt BEVs if UVAR is implemented. This is consistent with Priessner and colleagues' finding that non-car owners are among early EV adopters [54]. For UVAR, the finding highlights its potential to drive the uptake of sustainable transportation technology among non-drivers, who are likely to have different transportation needs and preferences compared to drivers. However, the results also raise the possibility of unintended consequences, such as the potential for increased motorisation, which may override some of the gains of UVAR implementation. The potential increase may imply the resurgence of transport problems such as congestion and safety concerns. A post-analysis assessment also found a significant association between commuting mode and possession of a driver's license. Most people who do not have a driver's license commute using public transport. Hence, suggesting the UVAR scenario may induce other transport users to find passenger car travel more attractive. Notwithstanding this, it is important to consider that the intention to adopt BEVs among non-drivers is likely shaped by multiple interrelated factors, including cost, availability, and perception of EVs, which were not investigated in this study. Regardless, this opens a window of opportunity for the further development of EV car-sharing or rental business models [54].

The factors identifying those likely to be neutral regarding BEV adoption add another interesting dimension to the study. First, it is acknowledged that respondents may opt for the neutral option (neither agree nor disagree) to convey uncertainty without conveying

ignorance or expressing a lack of opinion rather than a neutral opinion [55]. Yet the use of the midpoint cannot automatically be assumed to be a non-substantive response behaviour [56]. Building on this, the likelihood of those who are moderately concerned about parking issues to be neutral about BEV adoption is particularly insightful in the peculiar case of Budapest.

Parking concerns may not directly relate to the consideration of adopting BEVs. Notwithstanding, in the context of this study, perceived environmentally friendly vehicles, which include BEVs, registered within the country are currently not subjected to a parking charge within Budapest. This already incentivises the adoption of EVs generally [13,18]. The possibility of an UVAR scheme could add an extra nudge for potential adopters of BEVs who, under the scheme, would not need to cruise for an extended period for parking as is often experienced in the urban downtown area due to the potential reduction in vehicle usage. However, since the parking benefits are not exclusive to BEVs, respondents may be drawn to adopt PHEVs, which are often cheaper upfront than BEVs, albeit without the environmental benefits [57]. The sustainability of the non-financial incentives, such as free parking and an increased sensitisation on the ecological impacts of different modes of transportation, is crucial in promoting and maintaining sustainable mobility behaviours and choices. Although, a gradual phasing out of similar incentives for PHEVs and BEVs might be necessary.

Pro-environmental behaviours or attitudes might not predispose people to adopt BEVs even in light of the environmental gains the technology offers. This appears counterintuitive as many research findings have found a connection between pro-environmentalism and BEV adoption (e.g., [58,59]). Increased environmental consciousness encourages individuals to adopt battery electric vehicles as this awareness can enhance their sense of environmental responsibility and encourage them to adopt environmentally conscious behaviour. However, the direction of the model estimates for the environmentally conscious trip planning factor and the non-significance of the willingness to support UVAR measures suggest otherwise. BEVs may be associated with negative environmental impacts from the vehicle life-cycle perspective, including well-to-wheel emissions, sustainability issues with the extraction of raw materials, and recycling of key components at end-of-life [60,61]. These are particularly important within the Hungarian context. Only about 12% of electricity was produced from renewable energy sources in 2020, although this represents a 67% increase from 2010 data [62]. Wider adoption of BEVs within this context would only transfer emissions from urban areas. Furthermore, concerns about the potential negative impacts of a planned battery plant in Hungary on water sources and the environment can attribute a negative connotation to BEVs [63]. Beyond these concerns and the plausible explanations of cost barriers, access to charging infrastructure, and preference for other low-emission vehicles due to similar incentives, environmental consciousness identifies more with active mobility and public transport—the primary modes for promoting the sustainable mobility paradigm in urban areas [64]. The technological efficiency offered by BEVs in decarbonising transport is therefore expected to supplement other avoid and shift strategies such as improvement in accessibility, promotion of active mobility, public transport service improvements, and land-use reforms.

## 6. Conclusions

This study aimed to investigate the willingness of urban residents in Budapest, Hungary, to adopt BEVs if an UVAR is implemented and to determine the categories of urban commuters who would support such car-free policy measures. This study found a simple majority of respondents willing to adopt BEV following the implementation of vehicle access regulations in Budapest. Potential adopters of battery electric vehicles (BEVs) are typically middle-aged individuals who do not currently own a car, exhibit low sensitivity to travel costs during trip planning, and do not express significant concern regarding traffic congestion within the city. The results showed that age is important in determining an individual's willingness to adopt BEVs with younger and middle-aged individuals being

more likely to adopt the technology. Similarly, people who do not have valid driver's licenses and are predominantly public transport users may also be more willing to adopt BEVs if UVAR is implemented, which could be an avenue for unintended consequences.

On the other hand, this study could not establish that those who are conscious of the environmental impacts of their trips and who are supportive of UVAR are more likely to express an intention to adopt BEVs. These findings provide important insights for policymakers and stakeholders in the EV industry. It suggests that age-specific strategies, sustenance of non-financial incentives such as free parking for BEVs, and public campaigns on the environmental impacts of conventionally fuelled vehicles may effectively strengthen the potential effects of UVAR on BEV adoption.

This study has its limitations. It is important to note that these findings are based on a cross-sectional analysis and may not represent long-term trends or changes in behaviour. A possible selection bias resulting from recruiting participants online might limit the generalisability of the findings, especially since some of the groups are underrepresented. More cases will be required to develop more nuanced dependent variables for future modelling. In addition, due to the UVAR planning perspective of the dataset, some potentially influential factors are not considered in the present study, such as the electric vehicle technology perception and perceived benefits of BEVs. Further research is needed to better understand these variables' underlying mechanisms and interactions concerning BEV uptake and UVAR implementation.

**Author Contributions:** Conceptualisation, G.A.O. and F.M.; methodology, G.A.O. and F.M.; software and formal analysis, G.A.O.; writing—original draft preparation, G.A.O.; writing—review and editing, G.A.O. and F.M.; supervision, F.M. All authors have read and agreed to the published version of the manuscript.

**Funding:** This research received no external funding.

**Data Availability Statement:** Not applicable.

**Acknowledgments:** This research was supported by OTKA-K20-134760-Heterogeneity in user preferences and its impact on transport project appraisal led by Adam TOROK.

**Conflicts of Interest:** The authors declare no conflict of interest.

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
