# Peer review of "Investigating the Role of Urban Vehicle Access Regulations as a Policy Tool for Promoting Electric Mobility in Budapest"

_urbansci, doi:10.3390/urbansci7020039_

Round 1
Reviewer 1 Report
The proposed paper addresses a topic of a large interest at least for policy makers and for citizens.
How to influence the urban inhabitant’s availability to buy/use and electric vehicle instead of fuel-based vehicle is still a large debate.
The paper tries to enter into this debate to find the Budapest inhabitants propensity for electric vehicle in case that a restrictive access measure will be taken.
However, there are several shortcomings in the paper, which need more attention from the authors’ side, as follows:
1. The title of the paper has not sufficient clarity. It needs to suggest what is inside (that is in fact, an investigation in Budapest on the availability to the electric vehicle use/buy in case of access restriction for fuel vehicle)
2. The paper does not clearly and exactly reveal its novelty beyond the current state-of-the-art on this type of investigation against the actual literature. What is new and useful here beyond of the already used methodology for this type of research?
3. The presented research questions are not clearly related/connected to the survey’ questions.
4. There is a missing important discussion/analysis related to the inhabitants availability to accept an access restriction (and how large is this availability in the selected city) without an alternative of a very good quality services of public transportation.
5. The availability to use/buy an electric vehicle in case of restrictive access could be different for different battery electric vehicle: car, bicycle or other kind of electric two-wheals vehicle, but analysis is focussed only on electric cars .
6. The methodology need to be explained in more details, on the following issues:
-What mean of answers’ registration was chosen, and for what reason. The mean of investigation is important for the soundness of the proposed methodology.
-How the sample dimension was established and considered, in order to ensure the relevance of the performed study.
- How much that used dichotomy (by excluding Neutral response) might influence the results of study. Here, some relevant references related to the possible improvements of stated preferences analysis by dropping neutral answer should be mentioned.
-Possibility to use this survey as a pilot one in order to make adjustments for a future real and larger investigation.
Finally, please erase the inappropriate statement in lines 276-277.
Reviewer 2 Report
Dear authors,
The paper examines the willingness to adopt EVs based on perceptions of vehicle access regulations in Hungary. The topic is timely and relevant. The paper relies on a survey with a relatively small sample size (311 accepted respondents). Considering the difficulty of getting people to answer surveys the low number is acceptable and appears sufficient for the study.
Abstract:
Put more on focus on method and results rather than background.
Since you have stated two goals in the first sentence the second and third sentence (i.e. “This desired goal…”) should be rephrased to for example “The goals…”.
Introduction:
Line 27: The (t)ransport…
The first part of the introduction seem superfluous and could be shortened.
L 47-51 Sources?
Contextualize the study more in the Hungarian setting.
Clarify the purpose and the delimitation. Clarify what you mean with L 92 “However, in a hypothetical UVAR scenario.”
Preferably link the research questions to the literature review.
Materials and Methods:
Elaborate on how the survey was conducted. What do you mean exactly with self-administered survey? How were the surveys distributed (paper or digital?) and are there any other issues with overrepresentation, selection biases etc associated with how the survey was done? Currently these comments appear only in the conclusion (unless I missed something).
Motivate the choice of bivariate analysis. It seems rather limited in this case.
I do not really understand why the Neutral respondents were dropped. Wouldn’t they be interesting to examine since they might be more easily persuaded than the No group? Maybe provide a source that has made a similar choice?
Comment on the sample size and relate it to suggested minimum sample sizes for similar studies.
The income categories seem not to be suitable with almost 55 % in one category. Please comment on this in the method/discussion.
Contextualize the sample more by describing the Hungarian setting (population data, car ownership, share of EV ownership over time, incentives in place etc).
Please elaborate further on the analysis.
The methods chapter contain no sources. Please state sources that inspired your choices. For example age grouping etc. Also, if possible, provide further motives behind the decisions that you made.
Results
L. 138 This The..
How do you define micromobility?
Discussion
Is it really age or is it disposable income which you happened to not be able to discern more closely? Maybe discuss this further.
Are there potentially more connections between this and other studies or policies?
Contextualize the study more in the Hungarian setting.
Conclusions
Clearly state that the purpose was fulfilled and answer the research questions.
Highlight the counterintuitive results.
General comments:
The text needs proofreading.
Sources seem to be placed randomly before or after a full stop.
Sources are sometimes given at the end of the sentence and sometimes at the start etc. This is confusing when considering the referencing system that the journal uses. It would be ideal if the sources were given at the end of each statement/sentence. Examples of this confusing use of sources is found on lines 80-86.
Reviewer 3 Report
The manuscript analysed the adoption of battery-electric vehicles upon implementing an urban vehicle access regulations measure in a hypothetical scenario. The authors used a self-administered survey and a binary logit model to analyse data.
The problem is well presented. However, the literature could be explored in more detail to clearly show the paper's contribution.
Make the objective and contribution of the paper more clear in the introduction after presenting the research questions.
The method should be more detailed. For example, what is the purpose of doing a chi-squared test? The results should be explored in more detail. Moreover, the literature should support the results.
Discussion: Why are the results preliminary?
Conclusion: Remove lines 276-277. Answer all research questions proposed in the introduction.
Minor issues:
- Proofreading the manuscript by an English native speaker is recommended.
- In addition, the quality of Figure 2 should be improved.
Round 2
Reviewer 1 Report
The paper has been improved and is now more in line with the journal's quality standards.
Reviewer 2 Report
The authors have addressed my concerns and improved the paper.
Reviewer 3 Report
The authors included all suggestions made by this reviewer. The quality of the revised version is good, and the results are interesting.